# Androgen Receptor in Hormone Receptor-Positive Breast Cancer

**DOI:** 10.3390/ijms25010476

**Published:** 2023-12-29

**Authors:** Ashfia Fatima Khan, Samaneh Karami, Anthony S. Peidl, Kacie D. Waiters, Mariam Funmi Babajide, Tasneem Bawa-Khalfe

**Affiliations:** Center for Nuclear Receptors & Cell Signaling, Department of Biology & Biochemistry, University of Houston, 3517 Cullen Blvd, SERC Bldg., Rm 3010, Houston, TX 77204-5056, USAkdwaiter@central.uh.edu (K.D.W.);

**Keywords:** androgen receptor 1, hormone-positive breast cancer 2, post-translational modification 3, SUMOylation 4, AR-targeting drugs 5, clinical trials 6

## Abstract

Breast cancer subtypes expressing hormone receptors (HR+ BCa) have a good prognosis and respond to first-line endocrine therapy (ET). However, the majority of HR+ BCa patients exhibit intrinsic or acquired ET resistance (ET-R) and rapid onset of incurable metastatic BCa. With the failure of conventional ET, limited targeted therapy exists for ET-R HR+ BCa patients. The androgen receptor (AR) in HR-negative BCa subtypes is emerging as an attractive alternative target for therapy. The AR drives Luminal AR (LAR) triple-negative breast cancer progression, and LAR patients consistently exhibit positive clinical benefits with AR antagonists in clinical trials. In contrast, the function of the AR in HR+ BCa is more conflicting. AR in HR+ BCa correlates with a favorable prognosis, and yet, the AR supports the development of ET-R BCa. While AR antagonists were ineffective, ongoing clinical trials with a selective AR modulator have shown promise for HR+ BCa patients. To understand the incongruent actions of ARs in HR+ BCa, the current review discusses how the structure and post-translational modification impact AR function. Additionally, completed and ongoing clinical trials with FDA-approved AR-targeting agents for BCa are presented. Finally, we identify promising investigational small molecules and chimera drugs for future HR+ BCa therapy.

## 1. Introduction

Breast cancer (BCa) accounts for the largest rates of cancer-related mortalities among women. Despite notable advancements in reducing BCa mortality, the annual projection still stands at around 2 million new cases, resulting in over 650,000 reported global deaths each year [1]. Clearly, BCa remains an unmet therapeutic challenge. The four major BCa subtypes include Luminal A, Luminal B, human epidermal growth factor receptor 2 (HER2)-enriched, and triple-negative breast cancer (TNBC) [2]. Luminal A is the most frequently diagnosed (50%) BCa subtype and is associated with elevated levels of estrogen receptor (ER). Luminal B and a small fraction of HER2-enriched BCa subtypes also express ER. The majority of ER-positive BCa concurrently express the progesterone receptor (PR) as PR is an ER target gene. Collectively ER+/PR+ BCa are commonly classified as hormone receptor-positive (HR+) BCa [3]. In HR+ BCa, activation of ER drives tumor formation and disease progression [4]. Specifically, estrogen-activated ER initiates gene transcriptional programming that supports the aberrant proliferation of HR+ BCa cells [4]. Hence, the gold-standard treatment exploits the reliance of HR+ BCa on estrogen/ER signaling with anti-estrogen therapy or endocrine therapy (ET) [5]. Luminal A and B HR+ BCa have a good prognosis and response to ET. However, 15–20% of HR+ BCa patients are intrinsically resistant and 30–40% acquire resistance to ET. Patients with intrinsic or acquired ET-resistant (ET-R) BCa exhibit a more aggressive cancer with reduced disease-free survival and greater incidence of metastatic disease [6]. Limited targeted therapies currently exist for this large population of patients with ET-R HR+ BCa.

## 2. AR in Breast Cancer

The androgen receptor (AR) plays a critical role in normal breast physiology and breast cancer pathology. Approximately 70–90% of HR+ BCa, 30–60% of HER2+, and 20–30% of TNBC express ARs [7]. As TNBC is an aggressive BCa subtype with limited options for targeted therapy [8,9], AR serves as an attractive therapeutic target for the treatment of TNBC. The presence or absence of the AR in TNBC directly impacts disease outcomes. AR-negative, EGFR-positive breast TNBC breast cancer patients constitute the high-risk group, with highly proliferative tumors and consistently worse patient outcomes. Chemotherapy remains the mainstay for AR-negative TNBC patients. In contrast, 10–20% of all diagnosed TNBC patients are AR-positive; this TNBC subtype is commonly called Luminal Androgen Receptor (LAR) TNBC. LAR TNBC tumors exhibit a lower rate of proliferation, and LAR TNBC consistently has a better prognosis. In addition, AR and anti-androgen therapy serve as additional arsenal against the LAR TNBC tumor. As discussed later in this review, several clinical trials on AR+ TNBC patients with second-generation AR antagonists have shown promising results (NCT00468715).

The HR+ BCa subtype includes substantially higher levels of AR as compared to the other BCa subtypes. However, the role of ARs in HR+ BCa disease progression is still unclear. In the early stages of HR+ BCa, the ratio of AR: ER is used as a predictive marker for prognosis and clinical outcome following first-line endocrine therapy. The expression of high ARs to ERs in HR+ BCa patients is usually associated with good prognoses, with an enhanced treatment response in HR+ BCa patients [10]. Some studies show that in HR+ BCa, the AR engages in competition with Erα for the attachment to estrogen-responsive elements (EREs), resulting in impaired Erα transcription and the initiation of apoptosis [2]. However, in advanced stages of HR+ BCa, the expression of high ARs in ET-treated patients shows reduced distant metastatic free survival (DMSF) and worse prognoses [11]. This incongruency in AR function in HR+ BCa may be associated with its level of expression and/or the stage of disease; the exact consequence remains to be defined in breast cancer.

## 3. Genomic AR Activity in HR+ Breast Cancer

In the absence of ligands, the AR is inactive in the cytoplasm and is stabilized by association with chaperone proteins such as HSPs [12]. Specifically, the AR associates with Hsp70 and Hsp90. Hsp70 and Hsp90 are ATP-dependent proteins with canonical chaperone functions involved in the folding and degradation of client proteins; Hsp70 serves as a co-chaperone to Hsp90 [12]. The AR-Hsp70-Hsp90 maintains the AR in the inactive state in the cytoplasm while androgen binding causes the dissociation of ARs from the HSP complex [13]. In BCa, Hsp70 expression is shown to be upregulated and active in tumorigenesis and metastasis [14,15,16,17]. Hsp90 across multiple BCa subtypes (including HER2+ and HR+) also correlates with poor prognosis [18,19,20,21,22]. Hsp90 directly functions as a co-regulator for HER2 and HER2 signaling components [23,24] and therefore serves as a potential therapeutic target in BCa. Consistently, Hsp90 inhibitors have been developed and show positive results in preclinical models including BCa models. However, current Hsp90 inhibitors have not received FDA approval, citing off-target effects and lack of specificity [25,26,27]. The AR also interacts with Hsp27. Hsp27 is a small ATP-independent chaperone protein with its activity regulated through PTMs, particularly phosphorylation and SUMO modification [11,28,29,30]. Recently, we demonstrated that Hsp27 is a SUMO E3-ligase for AR in HR+ BCa as discussed below. Hsp27 has been shown to be highly expressed in BCa, inhibits apoptosis, and contributes to chemotherapy resistance and the epithelial–mesenchymal transition of BCa stem cells. Recent studies have shown a model of Hsp27 localization in the nucleus with the AR upon activation of the AR by dihydrotestosterone (DHT), with Hsp27 phosphorylation enhancing this localization event in molecular apocrine BCa [31].

Upon activation with ligands (dihydrotestosterone or DHT), the AR dissociates from its chaperones, homodimerizes, and is transported into the nucleus [12]. It binds to the androgen response elements (AREs), which are palindromic sequences on the promotor and enhancer regions of the DNA. The AR then associates with coregulatory proteins which can either enhance or repress gene transcription [32,33,34,35]. The interaction of ARs with coregulatory proteins is critical in AR signaling [34]. A subpopulation of coregulators has been studied in BCa. ARA70 (NCOA4) is a coactivator of ARs in HR+ BCa cells which synergizes AR activity and downregulates ER signaling [33]. Additionally, in TNBC, FOXA1 collaborates with ARs and supports the global transcription of AR-regulated genes. FOXA1 upregulation has been observed in HR+ BCa [35], suggesting an induction of AR signaling. The AR competitively sequesters ER coregulators p300 and SRC-3 in HR+ BCa, thus blocking ER activity [32]. Collectively, these coregulators dictate AR function in BCa. Further studies are necessary to establish the relationship between AR coregulators and their impact on cancer progression in HR+ BCa.

## 4. AR Structure

The AR gene is located on the X-chromosome and spans 2757 nucleotides across 8 exons [36]. The full-length AR contains 919 amino acids with a size of 110 kDa and contains three major structural domains: (1) the N-terminal domain (NTD), (2) the DNA-binding domain (DBD), (3) and the ligand-binding domain (LBD).

### 4.1. N-Terminal Domain (NTD)

The NTD comprises amino acids 1–555 (Figure 1A). While many studies outlining biochemically determined structures for the DBD and LBD exist, no confirmed crystal structure for the NTD has been determined. The NTD contains many disordered regions [37], leading to difficulties in stabilizing recombinant proteins to determine structure through crystallography. The NTD contains activation function-1 (AF1) (residues 142–485), Tau1 (110–370), and Tau5 (360–485) (Figure 1A) that are critical for full-length AR transcriptional activity [38]. The AF1 is a constitutively active domain that binds to transcription factors such as TBP and TFIIF [39]. The NTD contains a conserved consensus motif (FXXLF) that binds the coregulator/coactivators involved in transcriptional regulation [40]. This motif is also critical in intramolecular interactions with the carboxy terminus in the LBD. This interaction (called N/C interaction) that is mediated through the Tau1 and Tau5 domains within the AF1 makes the receptor fully transcriptionally active [41,42].

### 4.2. DNA-Binding Domain (DBD)

DBD consists of amino acids 556–623 and is directly bound to a hinge region that connects the DBD to LBD (Figure 1A,B). The DBD is a cysteine-rich region containing zinc-finger cores which are crucial for the recognition of specific DNA consensus sequences [43]. These sequences ensure AR binding to promoter and enhancer regions of genes to direct transcription. The DBD region is highly conserved among other hormone receptors [44]. The nuclear import of ARs is recognized by the nuclear localization signal (NLS), located within the hinge region bound between the DBD and LBD. Recent studies have shown that the binding of ligands exposes the NLS and allows for cofactor binding to translocate ARs to the nucleus and becomes subject to post-translational modifications (PTMs) [45,46].

### 4.3. Ligand-Binding Domain (LBD)

The LBD is located on the C-terminus of the AR and comprises amino acids 665–919 (Figure 1A,C). The LBD contains 11 α-helices and 4 ß-strands that fold into anti-parallel sheets, forming what is colloquially known as an α-helical “sandwich” [47]. AR LBD lacks the helix 2 that is present in the α-helical “sandwich” of other nuclear receptors; the AR’s α-helix nomenclature reflects the 12 helices present in the other nuclear receptor family members [47,48]. Some of these α-helices are critical for forming binding domains for coregulators and some classes of drugs to regulate transcription; namely the AF2 domain (Figure 1D,E). Helices 3, 5, and 12 form the hydrophobic pocket that makes up the AF2 domain [49]. Upon the ligand activation of ARs, the LBD conforms to expose the AF2 domain, making it accessible for coregulator proteins or chaperones that contain the present binding consensus motifs (FXXLF or LXXLL) to bind and regulate activity [50,51]. The LBD has been shown to bind coregulators that contain either FXXLF or LXXLL motifs; several studies indicate a preference for FXXLF-containing coregulators supported with the greater stability of an N-terminus to C-terminus interaction [52,53]. This LXXLL motif present on the LBD has been shown to bind through intramolecular protein–protein interactions with the FXXLF motif in the NTD [53,54], making the receptor transcriptionally active [42]. AR ligands and AR-targeting drugs bind to the ligand binding pocket located between helices 3, 5, 11, and 12 [47,48] (Figure 1F). As shown in Figure 1F, different classes of drugs and ligands bind within this pocket; antagonists (i.e., bicalutamide, darolutamide, apalutamide, R1881) and agonists (i.e., DHT, testosterone) bind with similar orientations and do not distort conformations of the helices relative to one another (Figure 1F).

## 5. AR Post-Translational Modifications

In addition to conventional androgen-dependent AR activity, post-translational modifications also direct AR signaling both in the presence and absence of AR ligands. PTMs play a crucial role in the regulation of ARs. The AR is subject to numerous PTMs; this review focuses on the modification of ARs in BCa, specifically AR phosphorylation, ubiquitination, and SUMOylation.

### 5.1. AR Phosphorylation

Phosphorylation impacts AR–protein interactions, specifically with protein interaction domains/motifs that reside near the AR phosphor acceptor site [55]. The AR full-length protein includes 16 serine, threonine, or tyrosine phosphorylation sites (Figure 1): S16, S81, S94, S213, S256, Y267, T282, S293, S308, Y363, Y424, S515, Y534, S578, S650, and S791 [56,57,58]. Most of the phosphor-site resides on the AR NTD. The phosphorylation events in ARs are regulated by site-specific kinases (including AKT, PMI1, etc.) which may or may not require androgen stimulation. AR phosphorylation at S16, S81, S256, S308, S424, and S650 increases in the presence of androgen activation, while S94 is constitutively phosphorylated in prostate cancer (PCa) cells [56]. Several studies on PCa indicate AR phosphorylation regulates both the genomic and non-genomic AR activity to drive cancer cell proliferation, survival, and invasiveness [59]. Deregulation of AR phosphorylation at S81, S213, and S650 is common in the HR+ BCa subtype, as evidenced in immunohistochemistry analyses of benign breast tissue, breast tumors, and metastatic breast tissue.

A 2-fold increase in nuclear and 1.7-fold increase in cytoplasm S213 was detected in breast cancer compared to the benign control, suggesting S213 may alter gene expression and the non-genomic role in cytoplasm to promote BCa progression. In contrast, nuclear and cytoplasmic S650 expression were both significantly reduced in BCa compared to benign breast tissue by 1.9-fold and 1.7-fold (*p* < 0.0001), respectively [60]. A comparative analysis shows high levels of phosopho-S213 AR in HR+ metastatic BCa tissue and invasive ductal carcinoma as well as HR- BCa [60]. Akt and PIM1 have been associated with AR phosphorylation at S231 in PCa [61,62]. Akt-mediated S213 phosphorylation in LNCaP cells show AR activation and cell survival. PIM-1S mediated S213 phosphorylation enhances AR proteolysis via MDM2 recruitment. However, the relationship between these kinases and ARs in HR+ BCa has not been studied. Similarly, AR S650 phosphorylation is higher in BCa metastatic sites, as validated using immunohistochemistry and Western blot analyses [60]. Collectively, these studies support elevated phospho-AR levels as a biomarker for BCa aggressiveness. In addition, in HR+ BCa lines, AR S81 phosphorylation increases following treatment with enobosarm (GTx-024), a selective AR modulator (SARM). HR+ patient-derived xenograft (PDX) HCI-13 tumors analyzed with Reverse Phase Protein Array (RPPA) following enobosarm (GTx-024) treatment reveal elevated levels of phosphorylated ARs [63]. Elevated phosphorylated S81 occurs with enobosarm (GTx-024) treated HCI-13 PDX and supports the need for AR S81 phosphorylation for AR activation and chromatin binding [63]. Enobosarm is currently in the clinical trial for HR+ BCa (NCT04869943). Future research should focus on the contribution of the other AR phosphorylation sites and their role in BCa progression. 

### 5.2. AR Ubiquitination

Ubiquitination is a dynamic and reversible PTM process in which ubiquitin covalently binds a lysine residue on a target protein. Accepting lysine residues on the target protein can include a single ubiquitin molecule or ubiquitin poly-chains; ubiquitin forms chains through internal lysine residues, specifically, K6, K27, K11, K29, K33, K48, and K63 [64]. Ubiquitin-specific enzymes ensure the conjugation process or ubiquitination of protein substrates; ubiquitination requires ubiquitin-activating enzymes (Ub-E1), ubiquitin-conjugating enzymes (Ub-E2), and ubiquitin ligases (Ub-E3). Ubiquitin conjugation directs the protein for proteasome-mediated degradation or impacts protein function. The AR is a substrate for ubiquitination with mass spectrometry analysis identifying K845 and K847 on the ligand binding domain (LTD) as the ubiquitination site (Figure 1) [64]. In PCa cells, the androgen activation of ARs initiates K48-linked poly-ubiquitin tags, 26S proteasomal degradation, and thereby loss of AR signaling [65]. In the same system, CHIP/STUB1, MDM2, SIAH2 function as Ub-E3 to support K48-ubiquitin linkages on AR and its subsequent degradation [66,67]. In contrast, Ub-E3 RNF6 facilitates K6, K27, or K63-linked poly-ubiquitin chains on ARs. These RNF6-driven poly-ubiquitin chains do not support AR degradation but instead serve as a scaffold for the recruitment of co-activators and the induction of AR genomic activity in PCa [64]. Substantially less is known about AR ubiquitination in BCa. However, reduced AR ubiquitination favors the induction of AR proteins in ET-R HR+ BCa cells [11]. AR ubiquitination in the same BCa line is dependent upon the SUMO PTM; SUMO, which commonly accumulates in ET-R BCa, decreases AR ubiquitination and subsequent proteasomal degradation of the AR protein [11]. Interestingly, ARs direct the transcription of select Ub-E3 in AR-positive TNBC. A report highlights that Ub-E3 FBW7 and MDM2 are downstream AR target genes. Anti-androgens induce FBW7 and MDM2 transcripts and promote the degradation of a target ion channel KCa1.1; ARs will likely also be subject to ubiquitin-mediated degradation as it is a target for the Ub-E3 activity of MDM2 in prostate cancer [68]. The relationship between ARs and the over 600 Ub-E3 families continues to be an interesting and evolving research area. For example, the interaction of ARs with other Ub-E3 like SIAH2, RNF6, and CHIP is undefined in BCa. Yet, a loss of nuclear Ub-E3 CHIP has been reported in ER+ MCF7, TNBC, and HER+ BCa [69]. Could a reduced CHIP expression then support the accumulation of ARs in BCa cells?

The ubiquitin PTM can be reversed through the actions of a class of cysteine proteases called deubiquitinases (DUBs). The AR interacts with and is deubiquitinated via USP26, USP14, USP12, USP10, and USP7, as reported in PCa [70,71]. In TNBC, USP14 deubiquitinates ARs and confers stability to ARs [71]. Furthermore, pharmacological targeting of USP14 with inhibitor IU1 or knockdown using shRNA stabilizes ARs, enhances the sensitivity of HR+ BCa cells to AR antagonist Enzalutamide, and, in turn, enhances TNBC cell apoptosis [72].

### 5.3. AR SUMOylation

The Small Ubiquitin-like Modifier (SUMO) also targets ARs [73]. SUMO PTM or SUMOylation requires a SUMO isoform (SUMO1, SUMO2, or SUMO3) to bind a lysine residue frequently within a conserved SUMO consensus site that includes the “(V/L/I)–K–x–E” motif. Like ubiquitination, the activity of the SUMO-specific family of E1-activating, E2-conjugating, and E3-ligase enzymes ensures the addition of SUMO to a target protein. For the AR, K386 and K520 on the NTD are part of a canonical consensus motif and serve as the predominant SUMO-acceptor sites [74]. In PCa cells, SUMO E3-ligase PIAS1 initiates AR SUMOylation; AR SUMOylation requires 1) overexpression of PIAS1 and androgen stimulation for a minimum of 15 min [74,75]. In this PCa system, PIAS1-mediated AR SUMOylation suppresses AR genomic activity [76]. In contrast, in HR+ BCa, particularly in ET-R HR+ BCa, the SUMO-modified AR is constitutively active and drives AR genomic activity. Recently, we identified a novel SUMO E3-ligase HSP27 that targets ARs for SUMOylation in ET-R HR+ BCa [11]. While HSP27 is a well-defined AR chaperone, limited substrates have been identified for its E3-ligase activity to date. HSP27 interacts with and supports the accumulation of SUMOylated AR in ET-R BCa, not PIAS1.

Deconjugating the SUMO-specific family of proteases (SENP) ensures the reversibility of SUMOylated substrates. SENP isoforms include SENP1, SENP2, SENP3, SENP6, and SENP7 [76,77]. The deSUMOylase enzyme SENP1 promotes AR deSUMOylation and maintains ARs in a transactivated state in PCa [78,79]. Furthermore, SENP1 expression is regulated by AR transactivation. An elevated level of SENP1 in prostate cancer cells has a significant impact on androgen-mediated cell growth, promoting the development and progression of prostate cancer [73,76,78,79]. While an equal level of SENP1 is expressed in observed in ET-S and ET-R BCa, an induction of global SUMOylation supports the SUMO-PTM of target proteins. The upregulation of SUMO isoforms coupled with HSP27 supports AR-SUMO PTM causing an accumulation of both modified and unmodified ARs. SUMO-modified ARs regulate epithelial–mesenchymal transition (EMT) gene transcription, suggesting its role in cancer progression and metastasis [11].

## 6. AR-Targeting Therapy for HR+ Breast Cancer Subtypes

Historically, androgens have served as therapeutic agents for the treatment of BCa. In the early 1940s, BCa patients were treated with androgen therapy, specifically testosterone propionate and fluoxymesterone, to counter the tumorigenic properties of estrogen [80,81]. While androgen therapy achieved positive clinical responses, the approach was virilizing with toxic side effects and was eventually replaced with anti-estrogen Tamoxifen. In place of hormones, current therapeutic strategies target ARs directly. However, incongruency in AR function exists among the BCa subtypes, and therefore, clinical trials equally focus on AR antagonists, modulators, and degraders. In this section of the review, we provide an overview of FDA-approved and investigational AR-targeting drugs and the ongoing clinical trials of these AR drugs for BCa (Table 1). 

### 6.1. AR Antagonists

First-generation competitive AR antagonist flutamide (Eulexin) disrupts androgen binding to ARs and thereby blocks AR genomic activity [82]. Flutamide was among the first AR antagonists approved for phase II clinical trials in metastatic BCa. Flutamide exhibited no significant antitumor activity in this patient cohort but instead caused gastrointestinal toxicities and various side effects in more than 50% of the participants [82]. Flutamide was replaced with newer, less toxic second-generation AR antagonists. In addition to blocking androgen binding ARs, second-generation AR antagonists bicalutamide (Casodex), enzalutamide (Xtandi), apalutamide (Erleada), and darolutamide (Nubeqa) prevent AR nuclear translocation. Bicalutamide (Casodex) has been tested in both TNBC and HR+ patients [9,83]. Bicalutamide treatment shows a 19% clinical benefit rate (CBR) for patients with AR-positive metastatic TNBC [9]. In contrast, phase II clinical trials with bicalutamide combined with an aromatase inhibitor (AI) show no positive benefit for HR+ BCa patients [83]. Another second-generation AR antagonist, enzalutamide, shows promise in pre-clinical mice xenografts with the inhibition of estradiol-driven MCF7 tumors. Notably, enzalutamide effectively suppresses dihydrotestosterone (DHT)-induced tumor growth in xenografts and promotes apoptosis in both HR+ (MCF7) and HR- (MDA-MB-453) cells [84,85]. However, like bicalutamide, enzalutamide did not show promise in clinical trials; a phase II clinical trial with 247 advanced or metastatic HR+ BCa patients showed that the combination of enzalutamide with the aromatase inhibitor exemestane does not significantly improve progression-free survival (PFS) compared to placebo. Clearly, there is a difference in response to AR antagonists in HR+ and TNBC patients and warrants a need to further assess why this discrepancy exists.

### 6.2. SARMS

Enobosarm is a nonsteroidal selective androgen receptor modulator (SARM) which has a tissue-specific androgenic activity [86]. The SARM presents more encouraging clinical outcomes for HR+ BCa than for patients with TNBC. A phase II trial with enobosarm for AR+ TNBC patients was terminated early due to a lack of drug efficacy (NCT02368691) [87]. In contrast, a phase II trial with 136 HR+ metastatic BCa patients has shown a good clinical benefit rate (CBR) with enobosarm monotherapy (NCT02463032) [88]. An 80% clinical benefit rate (CBR) is observed in patients with tumors that have an AR positivity of ≥40% as assessed by IHC; inversely patients with <40% AR positivity exhibits lower CBR (18%). Patients in the studies report minimal masculinization and virilizing effects with the enobosarm monotherapy. These positive findings serve as the foundation for a phase III randomized trial, ARTEST, currently in progress; the trial is designed to evaluate the effectiveness of enobosarm compared to ET in HR+ BCa patients with AR positivity ≥40% who progress after at least two lines of ET (NCT04869943). Another randomized phase III study comparing enobosarm in combination with the CDK4/6 inhibitor abemaciclib to standard second-line endocrine therapy is also underway (NCT05065411).

### 6.3. AR Degraders

Selective AR degraders (SARDs) include a class of AR-targeting drugs that effectively initiate AR protein degradation [89]. SARDs are currently in development to combat drug resistance in castration-resistant prostate cancer (CRPC). Compounds such as UT-155, UT-69, and UT-34, which are orally bioavailable, have been designed and synthesized through structural modifications of AR antagonists and enobosarm [89]. Several SARDs bind the AF-1 domain in the NTD to initiate ubiquitin-mediated proteasome degradation of ARs [89]. While preclinical studies of SARDs in prostate cancer have shown promising results, there is a lack of clinical evidence supporting their efficacy in PCa as well as BCa.

ARV-110 is an orally bioavailable Proteolysis Targeting Chimera (PROTAC) designed to harness the cell’s ubiquitin machinery, particularly Ub-E3, to degrade ARs [90]. Preclinical studies with ARV-110 are promising with its ability to degrade ARs in LNCaP, VCaP, and HR+ MCF7 cell lines at 1nM efficacy. In addition, the AR-PROTAC inhibits the proliferation of enzalutamide-resistant prostate cancer patient-derived xenograft (PDX) tumors [90,91]. Currently, ARV-110 is in phase II clinical trials for the treatment of metastatic CRPC, and based on the results from the NCT0388861 trial, it has shown promise as an effective treatment option for PCa patients [89].

## 7. Conclusions

The AR represents a promising biomarker and a therapeutic target for BCa patients. While AR+ TNBC patients show positive outcomes following anti-AR therapy, targeting ARs in HR+ BCa results show contradictory results. The conventional AR antagonist competes with the agonist to bind to the LBD of the AR, blocking its activity. As previously shown, drug/ligand binding on the LBD in the ligand binding pocket has marginal differences between conformational shifts in the alpha helices that stabilize the protein and the binding pocket. These drugs may be effective in the short term but acquired resistance to SARMs and other AR-targeting drugs remains a problem. The activity of ARs is regulated by PTMs, and these modifications can render AR-targeting drugs ineffective and can drive resistance. This loss of effectiveness could result from conformational shifts due to PTMs. For example, the SUMO-modified AR is constitutively active and does not respond to the antagonist Enzalutamide [11]; it remains to be determined if AR SUMOylation perturbs the enzalutamide binding site. Consistently degrading ARs may prove to be the best approach to target all forms of ARs (modified and unmodified). PROTACs hold potential as an effective approach to target ARs in HR+ BCa, particularly in cases of endocrine therapy-resistance.

## Figures and Tables

**Figure 1 ijms-25-00476-f001:**
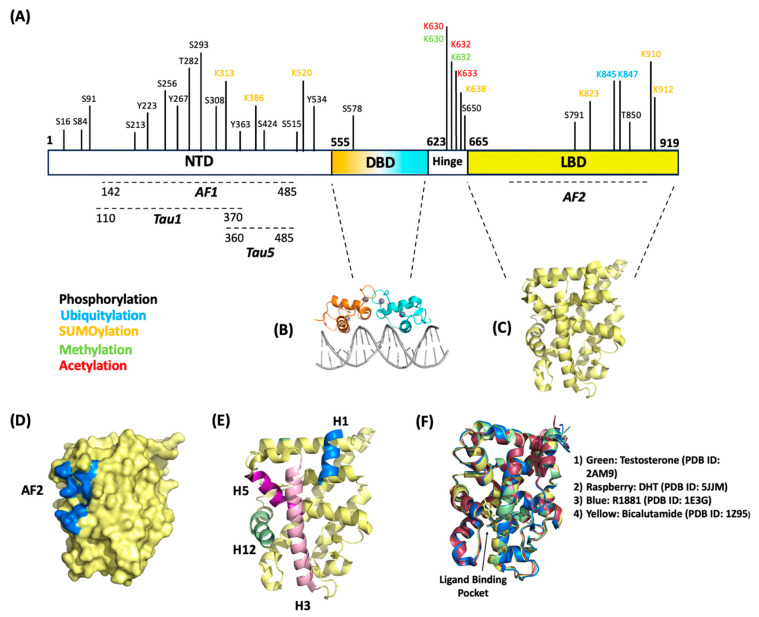
AR structure map. (**A**) AR linear map highlighting amino acid residues that are subject to post-translational modifications (phosphorylation, ubiquitination, and SUMOylation). Structural domains NTD, DBD, and LBD are shown here with their respective internal functional domains. (**B**) Crystal structure of the dimerized DBD bound to section of DNA (PDB ID: 1R4I). (**C**) Crystal structure of the LBD (PDB ID: 2PIW). (**D**) Surface representation of the LBD with AF2 domain represented in blue (PDB ID: 2PIW). (**E**) Ribbon representation of the LBD with helices 1, 3, 5, 11, and 12 emphasized. These helices are responsible for forming the hydrophobic pocket for the AF2 domain (PDB ID: 2PIW). (**F**) Structural overlay of AR LBD with different drugs/ligands bound in the ligand binding pocket.

**Table 1 ijms-25-00476-t001:** **AR-targeting drugs in clinical trials on HR+ BCa.** The table lists ongoing and completed clinical trials highlighting the study description of AR-targeting drugs in HR+ BCa.

NCT Number	Trial Name	Phase	Treatment interventions	Study Description	Status
NCT01597193	Safety Study of Enzalutamide (MDV3100) in Patients with Incurable BCa	Phase I	Enzalutamide ± Aromatase inhibitor/SERD	Phase I open-label, dose escalation study evaluating the safety, tolerability, and pharmacokinetics of enzalutamide in patients with incurable breast cancer.	Completed
NCT02953860	Fulvestrant Plus Enzalutamide in ER+/Her2- Advanced BCa	Phase II	Enzalutamide + Fulvestrant	Phase II study to evaluate the tolerability and clinical activity of adding enzalutamide to fulvestrant treatment in women with advanced BCa that are ER+ and/or PR+ and HER-2 normal.	Completed
NCT02007512	Efficacy and Safety Study of Enzalutamide in Combination with Exemestane in Patients with Advanced BCa	Phase II	Enzalutamide + Exemestane vs. Placebo + Exemestane	Phase II, randomized, double-blind, placebo-controlled, multicenter study of efficacy and safety of enzalutamide in combination with exemestane in patients with advanced BCa that is ER+, PR+, AR+ and HER2-normal.	Active, Not recruiting
NCT02955394	Preoperative Fulvestrant with or Without Enzalutamide in ER+/Her2- BCa	Phase II	Fulvestrant ± Enzalutamide	Randomized two-arm Phase II study to evaluate the efficacy of fulvestrant plus enzalutamide compared to single agent fulvestrant in postmenopausal women with locally advanced AR+/ER+/HER2- BCa who will have local surgery after ~4 months on treatment.	Active, Not recruiting
NCT03207529	Alpelisib and Enzalutamide in Treating Patients with Androgen Receptor and PTEN Positive Metastatic BCa	Phase II	Enzalutamide + Alpelisib	Phase IIb study of BYL719 (Alpelisib) in combination with Enzalutamide in Patients with AR+ and PTEN Positive Metastatic BCa	Recruiting
NCT02676986	Short-term Preoperative Treatment with Enzalutamide, Alone or in Combination with Exemestane in Primary BCa	Phase II	Enzalutamide ± Exemestane	Phase II Window of Opportunity Study of Short-term Preoperative Treatment with Enzalutamide (Alone or in Combination with Exemestane) in Patients with Primary ER+ BCa.	Unknown
NCT02910050	Bicalutamide Plus Aromatase Inhibitors in ER+/AR+/HER2- Metastatic BCa (BETTER)	Phase II	Bicalutamide + Aromatase Inhibitor	Phase II study to evaluate the efficacy and safety of bicalutamide and aromatase inhibitor in ER+/AR+/HER2-metastatic BCa patients with disease progression after treatment of an aromatase inhibitor.	Unknown
NCT04869943	Efficacy Evaluation of Enobosarm Monotherapy in Treatment of AR+/ER+/HER2- Metastatic BCa (ARTEST)	Phase II	Enobosarm	Phase II study to evaluate efficacy and safety of enobosarm monotherapy vs. active control for treatment of AR+/ER+/HER2- MBC With AR staining and previously treated w/nonsteroidal Aromatase Inhibitor, SERD & CDK 4/6 Inhibitor.	Active, Not recruiting
NCT05065411	Efficacy & Safety Evaluation of Enobosarm in Combination with Abemaciclib in Treatment of ER+HER2- Metastatic BCa (VERU-024)	Phase II	Enobosarm + Abemaciclib	Phase II study for Efficacy Evaluation of Enobosarm in Combination with Abemaciclib compared to Estrogen Blocking Agent for 2nd Line Treatment of ER+HER2- MBC in Patients who have shown previous disease progression on an estrogen blocking agent plus Palbociclib.	Recruiting
NCT01616758	Phase II Study of GTx024 in Women with Metastatic BCa	Phase II	Enobosarm	Phase II open label study to examine Androgen Receptor status and the activity of Enobosarm (GTx-024) in ER+ metastatic breast cancer patients who have previously responded to hormone therapy.	Competed
NCT02463032	Efficacy and Safety of GTx-024 in Patients with ER+/AR+ BCa	Phase II	Enobosarm	Phase II open label, multicenter, multinational, randomized, parallel design study to investigate the efficacy and safety of Enobosarm (GTx-024) on metastatic or locally advanced ER+/AR+ BCa in postmenopausal women.	Completed
NCT04142060	Targeting the PAM50 Her2-Enriched Phenotype with Enzalutamide in Hormone Receptor-Positive/Her2-Negative Metastatic BC (ARIANNA)	Phase II	Enzalutamide	Phase II open-label, non-randomized, two-cohort, multicenter, prospective study which evaluates the effect of enzalutamide on proliferation after 2 weeks (14-21 days) of treatment in patients with endocrine-resistant, locally advanced, or metastatic HR+/HER2-negative breast cancer.	Terminated

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
