# Peer review of "Androgen Receptor in Hormone Receptor-Positive Breast Cancer"

_ijms, 2023, doi:10.3390/ijms25010476_

Round 1

Reviewer 1 Report

Comments and Suggestions for Authors

In the review, the authors have summarized the existing knowledge on the relationship between AR and breast cancer, presenting a comprehensive survey of references. Covering such a expansive topic is challenging, but the review is well written to be beneficial for general readers. To enhance the precision and thoroughness of the paper, I recommend the following modifications. 

  1. L119-121, “In the absence of ligand, AR is inactive in the cytoplasm and is stabilized by association with chaperone proteins such as HSPs. Specifically, AR associates with Hsp70 and Hsp90. Hsp70 and Hsp90 are ATP-dependent proteins with canonical chaperone function involved in folding and degradation of client proteins; Hsp70 serves as a co-chaperone to Hsp90 [29].”: It seems that authors were supposed to cite reference29 for these sentences in L119-121 with adding [29] after the last sentences. Reviewer think that [29] should be after the first sentence or after the each sentence in L119-121. The same is true of L136-138, L241-244, L272-275, L279-282.

  1. L105-106: “The LBD contains 11 α-helices and 4 ß-strands that fold into anti-parallel sheets…”: The authors mentioned that "LBD contains 11 α-helices" but they also described Helices 12 in the same paragraph several times. This introduces an inconsistency in their explanation, which may be noticed by the readers.

  1. L93-94: “This motif is also critical in intramolecular interactions with a LXXLL motif in the LBD” and L111-112: “This LXXLL motif present on the LBD has been shown to bind through intramolecular protein-protein interactions with the FXXLF motif in the NTD”: The reviewer believes that the LXXLL motif is recognized as an interaction motif in coactivators. The AF2 region (helices 12) of the nuclear receptor can interact with the LXXLL (or FXXLF) motif of the coactivator. Additionally, the FXXLF motif in the NTD can interact with AF2. Just in case, the reviewer examined LXXLL in the AR LBD and identified one motif on H10. However, the Leu residues are not exposed; they are situated on the inside of the protein structure. As a result, interaction is not possible.

4.     Authors may consider adding the reference in the sentences on L113-114 “AR ligands and AR targeting drugs bind into the ligand binding pocket located between helices 3, 5, and 12 (Figure 1F).”. The reviewer has reservations about the composition of the LBP with these three helices and suggests supporting the claim with a relevant citation.

Author Response

Response have been uploaded as aPD

Reviewer 2 Report

Comments and Suggestions for Authors

This review discusses Androgen Receptor (AR) in hormone receptor positive breast cancer setting. It seems that the authors would like to highlight preclinical and clinical evidences demonstrating the benefits of AR therapy in HR+ BCa as well as discuss limitations of targeting AR in this subtype. However this review has several issues such as its poor organization and flow of information presented, constant back and forth between AR therapy in TNBC and HR+ breast cancer as well as prostate cancer.  For example: its unclear why there are separate subsections for the AR structure (Section 3), couldnt sections 3 and 5 been merged as AR biology to provide background on AR, followed by section 4 with separate sections/paragraphs of AR in such BCa subtype - for more clarity on how AR behaves in each subtype and this flow into section 6 describing AR therapy for BCa? Moreover, much of the information presented here has already been covered by other reviews (PMID: 31952272, 25722318, 36376977).

Comments on the Quality of English Language

Minor proofreading is required (eg: lines 165, 203)

Author Response

Response has been uploaded as a PDF.

Reviewer 3 Report

Comments and Suggestions for Authors

Serious attention has been drawn in this century to the role of androgen receptor (AR) in the induction and development of breast malignancies, in the prognosis for various forms and subtypes of breast cancer (BC) with both positive and negative hormone background. Methods of androgen targeting therapy are also the subject of thorough studies in many clinics. A very large number of investigations have been performed in this undoubtedly important area.

But despite the high intensity of the investigations of AR, the results of which are described in hundreds of interesting publications, many problems still remain unsolved.

The scientific aspect of the problem highlighted in the research of Khan A.F. et al and in their review is of great importance because this aspect describes the molecular basis of the AR function, the impact of the protein structure and its post-translation modification on this function.

The presented review is multifaceted. Besides the structural peculiarities of AR, it describes the details of AR targeting therapy in directly connection with the presented molecular basis of its function.

All   modern approaches to AR and anti-AR therapy of BC patients are reviewed, and references to clinical protocols for both in-progress and completed trials are also given in this manuscript.

The conclusions of this review, which fully correlate with the presented and discussed data, are highly thought-provoking for the further research aimed at solving the questions about the efficient use of AR as a biomarker and as a target for the therapy of BC patients.

The manuscript is fully consistent with the contents of the International Journal of Molecular Sciences and can be recommended for publication.

Author Response

We thank the reviewer for their comments.